# Different Methods for Seminal Plasma Removal and Sperm Selection on the Quality and Fertility of Collared Peccary Sperm

**DOI:** 10.3390/ani13121955

**Published:** 2023-06-11

**Authors:** Maria V. O. Santos, Andréia M. Silva, Leonardo V. C. Aquino, Lhara R. M. Oliveira, Samara S. J. Moreira, Moacir F. Oliveira, Alexandre R. Silva, Alexsandra F. Pereira

**Affiliations:** 1Laboratory of Animal Biotechnology, Federal Rural University of Semi-Arid, Mossoro 59625-900, RN, Brazil; maria.santos51976@alunos.ufersa.edu.br (M.V.O.S.); leonardo.aquino03297@alunos.ufersa.edu.br (L.V.C.A.); lhara.oliveira@alunos.ufersa.edu.br (L.R.M.O.); 2Laboratory of Animal Germplasm Conservation, Federal Rural University of Semi-Arid, Mossoro 59625-900, RN, Brazil; andreiam.silva@alunos.ufersa.edu.br (A.M.S.); samara.sandy@bol.com.br (S.S.J.M.); alexrs@ufersa.edu.br (A.R.S.); 3Laboratory of Applied Animal Morphophysiology, Federal Rural University of Semi-Arid, Mossoro 59625-900, RN, Brazil; moacir@ufersa.edu.br

**Keywords:** wild ungulates, density gradient centrifugation, swim-up, washing by centrifugation

## Abstract

**Simple Summary:**

The collared peccary is a wild ungulate important for the maintenance of ecosystems in the Americas. The in vitro fertilization (IVF) technique can contribute to the conservation of and knowledge about the reproductive characteristics of this species. The collared peccary’s spermatozoa are sensitive to stress conditions. Therefore, the steps of seminal plasma (SP) removal and sperm selection for fertilization need to be determined. Consequently, the most used methods in domestic ungulate species, namely, swim-up, Percoll^®^ gradient (PG), and washing by centrifugation (WC), were evaluated. Initially, it was observed that regardless of the sample preparation for the swim-up (with SP or without SP), this method was not efficient in isolating motile sperm. Different PG concentrations were compared, revealing PG 45–90% recovered spermatozoa with better motility patterns compared to PG 35–70% and non-selected sperm. Then, PG 45–90% was compared to WC for sperm motility and heterologous IVF with swine oocytes. PG 45–90% showed better motility patterns, but the rate of fertilization and development of embryos were similar compared to WC. Therefore, PG 45–90% and WC are recommended for SP removal and sperm selection in the collared peccary.

**Abstract:**

Methods for seminal plasma (SP) removal and the selection of collared peccary sperm for fertilization were compared. The experiments evaluated the following: the (I) impact of centrifugation for SP removal before swim-up for sperm selection and (II) a comparison of different Percoll^®^ gradient densities (PG 45–90% and PG 35–70%). Non-selected sperm served as the control. Sperm quality was assessed based on motility patterns, morphology, membrane functional integrity, viability, reactive oxygen species (ROS), glutathione (GSH), and DNA integrity. Subsequently, the most successful group in the previous experiment and washing by centrifugation (WC) were compared for motility patterns and fertilization using pig oocytes. Swim-up decreased motility and enhanced ROS compared to the control. Centrifugation before swim-up harmed integrity and viability compared to the control. PG 45–90% (96.8 vs. 69.7 vs. 40.7 µm/s) allowed for a better velocity average pathway (VAP), a better velocity straight line, and better linearity (LIN) than those of the control and PG 35–70% (88.4 vs. 56.0 vs. 27.3 µm/s). Thus, PG 45–90% was used for fertilization. PG 45–90% obtained a higher VAP, a higher amplitude of the lateral head, straightness, and higher LIN than those of the control and WC. Cleavage (25.2–26.3%) and morula (8.1–10.5%) rates did not differ between the groups. Therefore, PG 45–90% and WC were efficient in isolating collared peccary sperm capable of fertilizing pig oocytes.

## 1. Introduction

The collared peccary is a wild ungulate, internationally categorized as a least-concern species in terms of the risk of extinction [1]. However, its population in the Caatinga and Atlantic Forest biomes has been declining due to habitat lost and predatory hunting [2]. This population loss affects the ecological role of the collared peccary, which involves seed dispersal [3] and maintaining the balance of the food chain for large carnivores [4]. However, collared peccaries adapt well to captive breeding, rendering them suitable candidates for the Assisted Reproduction Techniques (ARTs) performed from the view of scientific and commercial development [5].

Different strategies were reported for developing semen technologies and physiological characterization of the semen of collared peccaries. This report highlights advances in semen collection [6], evaluation [7], refrigeration [8], and cryopreservation [9,10] as well as the protein [11] and biochemical [12] characterization of semen and artificial insemination [13]. However, an effective method for seminal plasma (SP) removal and sperm selection has not been established for collared peccaries. While there are many studies on pigs and ruminants, the ART protocols used in domestic species cannot be applied to wild animals without prior evaluation [14].

In general, the SP removal and sperm selection from semen samples can benefit ARTs, including cryopreservation followed by in vitro embryo production (IVEP) [15]. Specifically, to perform IVEP using in vitro fertilization (IVF) and intracytoplasmic sperm injection (ICSI), the separation of sperm from SP or extenders is crucial in any species [16]. This step allows for the isolation of sperm with better kinetic patterns and the removal of unwanted cells and microorganisms [17]. This is important, since the prolonged contact time among motile spermatozoa, immature spermatozoa, and cell debris may induce the production of reactive oxygen species (ROS) related to DNA damage [18].

Sperm samples collected from collared peccaries demonstrated sensitivity to high-intensity centrifugation (750× *g*) and stressful conditions such as cryopreservation [6,19]. Thus, the simple and economical swim-up method of sperm separation, where sperm swim above the surface of the medium, is suitable for sensitive sperm [20]. The sample used for swim-up selection may contain SP or it can be removed using low-intensity centrifugation. SP contains decapacitating factors that may impair the ability to achieve capacitation and fertilization according to exposure time; however, its removal by centrifugation can also sensitize collared peccary sperm before swimming [21,22,23].

The Percoll^®^ gradient (PG) technique was used for sperm separation to concentrate motile sperm and assess fertility-associated features in samples from several species, such as pigs [24], goats [24,25], and cattle [17,26]. However, Percoll^®^ has an endotoxic effect, so its use was banned in human ARTs [27]. In this technique, sperm traverse varying densities of Percoll^®^ to reach the bottom of the tube. The gradient of 45–90% Percoll^®^ is used more frequently for swine and ruminants, followed by the 35–70% gradient. The density of Percoll^®^ influences the sperm quality and the recovery rate; hence, the best conditions for the collared peccary need to be optimized, and the tolerance of the species to Percoll^®^ should be assessed [28,29].

In contrast, centrifugation washing (WC) is a simple and fast method that recovers a high sperm concentration, removes SP, and is widely used [25,30]. Furthermore, simple centrifugation was previously used for sperm processing in collared peccaries, although a low intensity was recommended [6].

Therefore, this study aimed to evaluate different methods for SP removal and sperm selection of collared peccaries. The effects of (i) centrifugation to remove SP before swim-up, (ii) two densities of PG for sperm separation, and (iii) the fertility of sperm separated by the established methods or WC using heterologous IVF with porcine oocytes were analyzed.

## 2. Materials and Methods

### 2.1. Bioethics and Chemicals

The experiments were performed following the rules of the Animal Ethics Committee of the Federal Rural University of the Semi-Arid Region (no. 30/2019) and Chico Mendes Institute for Biodiversity Conservation (ICMBio, no. 71834-1). Unless otherwise stated, the reagents were obtained from Sigma-Aldrich (St. Louis, MO, USA).

### 2.2. Semen Collection

Nineteen ejaculates were collected from sexually mature collared peccaries (25–30 months old). Animals were obtained from the Center of Multiplication of Wild Animals (CEMAS/UFERSA, Mossoró, RN, Brazil; 5°10′ S, 37°10′ W) and registered at the Brazilian Institute of Environment and Renewable Natural Resources (IBAMA, no. 1478912). The collared peccaries were housed in outdoor paddocks (20 × 3 m^2^), which included a covered area (3 × 3 m^2^) with access to water ad libitum, and were provided feed and fruits.

The animals were fasted for 12 h before semen collection, physically restrained using a hand net, and anesthetized using intravenously administered propofol (Propovan, Cristalia, Fortaleza, Brazil) given as a bolus (5 mg/kg) [31]. Subsequently, they were placed in the lateral decubitus position to collect semen using an electroejaculation protocol, previously optimized for collared peccaries [6]. Fresh ejaculates were immediately evaluated for appearance, color, pH, and concentration using a Neubauer chamber.

### 2.3. Experimental Design

The study was divided into three experiments, and in the control group (not separated from SP), semen was immediately evaluated after collection. Sperm were evaluated for motility patterns, morphology, membrane functional integrity, viability, recovery rate (concentration after selection with a Neubauer chamber), ROS and GSH levels, and DNA integrity. Moreover, heterologous IVF was performed using swine oocytes.

#### 2.3.1. Experiment I: Swim-Up

Sperm separation was evaluated using the swim-up method after applying centrifugation or without centrifugation to remove SP. Fresh semen from each ejaculation was divided into two parts. One part was diluted to a concentration of 100 × 10^6^ sperm/mL (non-centrifuged group) and subjected to swim-up. The other part was diluted in a 1:1 ratio and centrifuged at 300× *g* for 3 min, the pellet was resuspended, and the concentration was adjusted to 100 × 10^6^ sperm/mL and then subjected to swim-up. In total, seven animals were used, with each ejaculate considered as a replicate. Throughout the procedure, the Sperm Tyrode Lactate (SPTL) medium (100 mM NaCl, 3.1 mM KCl, 0.4 mM NaH_2_PO_4_, 21.6 mM Na lactate, 25 mM NaHCO_3_, 0.5 mM caffeine, 2 mM CaCl_2_, 1 mM MgCl_2_, 10 mM HEPES, 0.6% bovine serum albumin (BSA), 1 mM sodium pyruvate, and 1% antibiotic–antimycotic solution) was used. Separation was performed following the method reported by Olivares et al. [25], with modifications described below. Initially, 250 µL of the sperm suspension was placed in a 15 mL tube, and 1.0 mL of SPTL medium was carefully added to it. The swim-up step was performed for 1 h at 38.5 °C and 5% CO_2_, with tubes tilted at 45°. Subsequently, the supernatant (200–300 µL) was carefully removed and centrifuged (300× *g* for 3 min) to concentrate the sperm suspension. The separated sperm were resuspended in SPTL medium and evaluated.

#### 2.3.2. Experiment II: Percoll^®^ Gradient

Different Percoll^®^ gradient densities were compared, including 45–90% [28] and 35–70% [32], for the separation of collared peccary sperm. The gradient volume and centrifugation conditions were selected following previous experiments. Briefly, the total volumes of 1.0 and 2.0 mL of the Percoll^®^ gradient were compared, and better recovery was detected using a total volume of 1.0 mL (500 µL 45% and 500 µL 90% Percoll^®^). The recovery after centrifugation at 900× *g* for 15 and 30 min was evaluated. A long amount of time did not increase the recovery; therefore, 15 min was considered the optimum recovery time.

In a 1.5 mL conical microtube, 500 µL of 45% Percoll^®^ was added to the bottom, and, subsequently, 500 µL of Percoll^®^ 90% was carefully dispensed on it, while avoiding mixing both dilutions. The same procedure was performed to study the 35–70% concentration gradient. Fresh semen from each ejaculation was diluted to a concentration of 100 × 10^6^ sperm/mL using an SPTL medium. In total, six animals were used, and each ejaculate was considered as a replicate. A 250 µL sample of this sperm suspension was placed over the gradients and centrifuged at 900× *g* for 15 min (37 °C). The supernatant was discarded, and the pellet was resuspended in SPTL medium, which was subsequently centrifuged (300× *g* for 3 min) to remove Percoll^®^. The selected sperm were resuspended in SPTL and evaluated.

#### 2.3.3. Experiment III: Heterologous IVF

In terms of fertilization performance, the most efficient conditions for sperm separation observed in the previous experiments were compared to the washing by centrifugation (WC) method. To prevent cell damage, a low centrifugation speed and duration were used. Prior to the experiments, the efficacy of the protocol in forming pellets containing spermatozoa was confirmed. Thus, fresh semen was diluted using SPTL medium in a ratio of 1:1, and 1.0 mL of this suspension was subjected to centrifugation steps of 100× *g* for 3 min repeated three times [33]. After washing, the pellets were resuspended, and the concentration was adjusted for evaluation and IVF.

Sperm from fresh samples separated using various techniques were utilized for IVF with swine oocytes. Six animals were used, and each ejaculate served as a replicate. Sperm were evaluated for motility patterns, morphology, and effectiveness for oocyte–sperm interaction and embryonic development.

### 2.4. Sperm Evaluations

#### 2.4.1. Motility Patterns

The motility of sperm was analyzed using a computer-assisted sperm assessment system (IVOS 7.4 G; Hamilton-Thorne Research, Beverly, MA, USA), with settings validated for collared peccaries [8]. Five viewed fields were randomly selected and scanned. The instrument settings were as follows: temperature, 37 °C; 60 frames/s; 45 as minimum contrast; 30% straightness threshold; 10 μ/s low-velocity average pathway (VAP) cutoff; and 30 μ/s medium VAP cutoff. The total motility (%), progressive motility (%), VAP (μm/s), velocity straight line (VSL; μm/s), curvilinear velocity (VCL; μm/s), amplitude of the lateral head (ALH; μm), beat cross frequency (BCF; Hz), straightness (STR; %), and linearity (LIN; %) of sperm were analyzed. Sperm populations were subdivided into four categories: rapid, medium, slow, and static.

#### 2.4.2. Morphology and Functional Integrity of the Membrane

Sperm present in smears stained with Bengal rose (Cromato^®^) were viewed under an optical microscope (1000×; 100 cells per slide) [34], and their morphology was investigated. The functional integrity of the sperm membrane was reflected by the sperm osmotic response, which was evaluated using 100 sperm samples by a hypo-osmotic swelling test. For this purpose, an aliquot of sperm suspension was incubated in a hypo-osmotic solution (distilled water: 0 mOsm/L), and cells with a swollen curled tail were considered to have a functional membrane [7].

#### 2.4.3. Viability and Mitochondrial Activity

Sperm viability was analyzed based on the structural integrity of the plasma membrane and mitochondrial activity. For this evaluation, spermatozoa were incubated with 40 µg/mL Hoechst 33,342 at 37 °C for 10 min and with 0.5 mg/mL propidium iodide and 500 µM chloromethyl-X-rosamine for 8 min (Invitrogen, Carlsbad, CA, USA). Next, 100 cells were analyzed using an epifluorescence microscope (400×; Olympus BX51TF). Individual spermatozoon appearing blue with a midpiece appearing red was considered to have an intact sperm membrane and mitochondrial activity [35].

#### 2.4.4. Integrity of the DNA

To assess DNA integrity, an assay based on an acid challenge that denatures DNA molecules from a susceptible chromatin structure was used, which included DNA strand separation and intercalation of the acridine orange probe to DNA generating red (denatured single-stranded DNA) or green (double-stranded DNA) fluorescence [36]. Initially, sperm suspensions were smeared and dried in the air. The method reported by Martins et al. [37] was followed, with the exception that 25 min was required as the time to verify chromatin integrity. The smears were fixed in Carnoy’s solution (methanol and glacial acetic acid in a 3:1 ratio) for 24 h, dried again, and incubated for 25 min in a buffer solution composed of 15 mM Na_2_HPO_4_ and 80 mM citric acid (pH 2.5) at 75 °C to check chromatin stability. Smears were stained using acridine orange (0.2 mg/mL) for 10 s, washed with distilled water, and covered using a coverslip. A total of 100 cells were evaluated using epifluorescence microscopy. Sperm with green fluorescence on the head contained intact DNA, and those with orange or red fluorescence had denatured DNA.

#### 2.4.5. Oxidative Status

To assess the oxidative status of sperm cells, ROS levels were evaluated by tagging sperm with 2′,7′ dichlorodihydrofluorescein diacetate (H_2_DCFDA; 10 µM; Invitrogen, Carlsbad, CA, USA), and GSH levels were analyzed using 7-amino-4-chloromethylcoumarin (CellTracker Blue; 10 µM; Invitrogen, Carlsbad, CA, USA) [38]. Briefly, 2.5 mL of phosphate-buffered solution (PBS) was added to a 0.5 µL (0.1 M) aliquot of each fluorescent probe. Thereafter, 50 µL of each diluted probe was added to 50 µL of the sperm suspension at 40 × 10^6^ sperm/mL. The samples were then incubated at 37 °C for 30 min in the dark. Subsequently, the probes were removed by centrifugations (500× *g*/5 min) repeated twice, and the precipitate was resuspended in PBS. The slides were visualized using an epifluorescence microscope, and microscopic images were evaluated using ImageJ software. From each sample, up to 100 sperm samples were selected for the quantification of fluorescence intensity. Fresh spermatozoa were used to calibrate the measurements, and the measured value of each sperm was divided by the mean of the calibrator to generate relative expression levels (arbitrary fluorescence units (AFU)).

### 2.5. Heterologous IVF

#### 2.5.1. Oocyte Collection and In Vitro Maturation

Ovaries were collected from swine females in a local slaughterhouse, stored in NaCl (0.9% at 35–37 °C), and transported to the laboratory. Follicles with 3–6 mm diameter were aspirated using a 21 G needle and 5.0 mL syringe. Oocytes with more than one layer of cumulus cells and a homogeneous cytoplasm were selected. The in vitro maturation (IVM) was performed in drops (100 µL) covered with mineral oil in a controlled humid atmosphere at 38.5 °C and 5% CO_2_ for 42–44 h. The maturation medium was composed of TCM199 supplemented with 2.2 g/L sodium bicarbonate, 25 mM HEPES, 0.3 mM sodium pyruvate, 5 µg/mL myo-inositol, 10% porcine follicular fluid, 5 ng/mL epidermal growth factor (EGF), 20 µg/mL follicle-stimulating hormone associated with luteinizing hormone (FSH/LH; Pluset^®^, Hertape Calier, Juatuba, MG, Brazil), and 1% antibiotic–antimycotic solution [39].

#### 2.5.2. Co-Incubation and Embryo Development

After IVM, the oocytes were washed and pelleted a few times to remove excess cumulus cells. They were grouped (10–15 oocytes) and incubated at 38.5 °C and 5% CO_2_ with separated sperm in drops of 50 μL of IVF medium (114 mM NaCl, 3.2 mM KCl, 0.35 mM NaH_2_PO_4_, 10 mM Na lactate, 5 mM glucose, 25 mM NaHCO_3_, 2 mM caffeine, 4.7 mM CaCl_2_, 0.5 mM MgCl_2_, 10 mM HEPES, 0.3% BSA, 0.11 mM sodium pyruvate, and 1% antibiotic–antimycotic solution) [40], covered with mineral oil. The sperm concentration was 3 × 10^5^ sperm/mL, and the IVF process lasted for 6 h [41].

After the IVF process, the structures were washed and pipetted to remove the unbound sperm and cumulus cells. The presumed heterologous zygotes were subjected to in vitro development in 50 μL drops of synthetic oviduct fluid (SOF) supplemented with 0.2 mM sodium pyruvate, 0.2 mL L-glutamine, 0.34 mM citrate sodium, 2.8 mM myo-inositol, 2% essential amino acid solution, 1% non-essential amino acid solution, 1% antibiotic–antimycotic solution, 5 mg/mL BSA, and 2.5% fetal bovine serum [42]. After 48 h of culture, 50% of the medium was changed, and the non-cleaved cells were removed and evaluated for oocyte–sperm interaction and fertilization using Hoechst 33,342 (10 µg/mL, 15 min) under a fluorescence microscope. The cleaved embryos were quantified and classified according to the number of cells (two cells and >three cells) and cultured for 6 days. Embryos that reached the morula stage were quantified.

### 2.6. Statistical Analysis

Data were expressed as mean ± standard error (one male/one repetition) and were analyzed using GraphPad software (GraphPad Software Inc., La Jolla, CA, USA). All results were evaluated using Shapiro–Wilk test to verify the normality of residuals, and the homogeneity of variance was examined using the Levene test. Since the data did not follow a normal distribution, they were arcsine-transformed and analyzed using ANOVA followed by Tukey’s test. The chi-squared test was employed to compare data on sperm–oocyte interaction, fertilization, cleavage rate, and morula rate as independent categorical variables. Statistical significance was set at *p* < 0.05.

## 3. Results

All 19 ejaculates were white watery samples, with pH 7.0 and an average concentration of 450 ± 89.1 × 10^6^ sperm/mL. Only samples showing motility greater than 80% were used in the analyses.

### 3.1. Experiment I: Swim-Up

The swim-up method using centrifuged or non-centrifuged samples significantly decreased the values of motility, progressive motility, VAP, VSL, VCL, ALH, and rapid sperm frequency compared to that of the control. Moreover, swim-up recovered a higher percentage of static sperm (Table 1).

The percentage of cells with normal morphology was similar between groups (Figure 1A). The non-centrifuged group showed membrane functional integrity and viability similar to the control, whereas these parameters in the centrifuged group were significantly decreased (Figure 1B,C). Sperm recovery did not differ between centrifuged and non-centrifuged samples before separation (Figure 1D). ROS levels increased after the swim-up compared to the control, with the highest value observed in the centrifuged samples (Figure 2A; *p <* 0.05). GSH levels were higher (*p <* 0.05) in the centrifuged group (1.35 ± 0.10 AFU) than in the non-centrifuged group (0.98 ± 0.04 AFU) and control (1.00 ± 0.03 AFU) (Figure 2B). DNA integrity was similar in the sperm separated and control groups (Figure 2C). Hence, among the sample preparation methods, the non-centrifuged group caused less damage to the sperm quality. In general, the swim-up method did not isolate sperm with better motility and other quality-related parameters of collared peccary under the studied conditions. Therefore, this method was not used in our third experiment.

### 3.2. Experiment II: Percoll^®^ Gradient

The kinetic and motility parameters are listed in Table 2. The total and progressive motilities were similar between the control and PG 45–90% groups. However, the 35–70% concentration gradient showed a significant and sharp decrease in these parameters. The separation with PG 45–90% significantly isolated sperm with better VAP, VSL, and LIN compared to those of the control and PG 35–70%, which showed harmful effects on ALH, STR, and rapid, medium, and static subpopulations. Moreover, the 45–90% concentration gradient recovered a higher percentage of static sperm compared to the control, which was significantly more increased by the 35–70% concentration gradient (*p <* 0.05).

Normal morphology, membrane functional integrity, and viability were superior in the PG 45–90% and control groups compared to the PG 35–70% group, which impaired sperm quality (Figure 3A–C). However, the sperm recovery values for the two tested Percoll^®^ concentration gradients were similar (Figure 3D). The levels of ROS and DNA integrity were similar in all sperm (Figure 4A,B). Conversely, PG 35–70% significantly decreased the levels of GSH (0.55 ± 0.02 AFU) compared with the PG 45–90% (0.85 ± 0.03 AFU) and control (1.00 ± 0.04 AFU) groups (Figure 4C). Hence, the separation of sperm from collared peccary using PG 45–90% was used for heterologous IVF, since this gradient recovered sperm showing better kinetic parameters and maintained sperm quality.

### 3.3. Experiment III: Heterologous IVF

Sperm separated for heterologous IVF were evaluated for motility patterns and normal morphology (Table 3). The motility patterns of all sperm were similar in the control and WC groups. PG showed higher VAP, ALH, STR, and LIN values compared to the control and WC groups. Nevertheless, PG significantly recovered more static sperm compared with the control and WC groups. No significant difference was detected in the percentage of morphologically normal sperm.

The number of sperm bound to mature, not mature, and total oocytes did not differ with the various separation methods (Table 4). Furthermore, fertilization analyzed by the presence of the second polar body and two pronuclei did not differ between PG and WC, presenting fertilized matured oocytes at 22.5–25.0%.

The separation techniques did not alter the cleavage rate in fertilized oocytes, which was detected to be 26.3% and 25.2% in WC and PG, respectively (Figure 5A,B). More than 88% of the cleaved hybrid embryos had more than three cells. In total, 10.5% and 8.1% of embryos reached the morula stage in the WC and PG groups, respectively (Figure 5C,D). Among cleaved embryos, the frequencies of morula development achieved with WC and PG were 40% and 32.3%, respectively. No difference in morula development was found between the experimental groups (WC and PG).

## 4. Discussion

This study evaluated different separation techniques for the sperm of collared peccaries for the first time. The Percoll^®^ density gradient of 45–90% was identified as the most effective method to remove SP and select motile sperm from these animals. Although the total and progressive motilities were similar to those of the control, the quality of the sperm kinetics expressed in terms of VAP, VSL, and LIN was superior when PG 45–90% was compared to PG 35–70%. In swine sperm, these parameters were reported to be positively correlated with fertility [43,44,45].

Previously, Matás et al. [28] analyzed three density combinations of PG (45–60%, 60–75%, and 45–90%) for swine sperm, where 45–90% recovered morphologically normal sperm with less DNA damage and better motility and capacitation compared to the control and other PG densities. In the collared peccary, PG 45–90% maintained the sperm quality noticed in the control set and did not induce oxidative stress or impair DNA integrity. Therefore, Percoll^®^ did not show toxic effect to this species, despite its toxicity to human samples, which is why its use is not allowed in fertilization clinics because of the risk of contamination with endotoxins [46].

In contrast, sperm selection with PG 35–70% had adverse effects on the samples of collared peccaries. This suggests that the low-density gradient allows for a stronger impact of centrifugation on sperm, making them more sensitive to this process. It is possible that the pellet formed was denser and faster, causing the sperm cells to stay in contact for a longer time compared to PG 45–90%. Additionally, PG 35–70% might not effectively separate cell debris and dead cells. These may cause structural damage and oxidative stress that affect the quality and functionality of collared peccary sperm.

No significant increase in ROS levels was detected, whereas GSH levels decreased after the PG 35–70% separation process. Decreased levels of GSH, a main antioxidant defense component of sperm cells, indicate an imbalanced cellular defense that makes sperm more susceptible to damage from oxidative stress [47]. In contrast, PG 35–70% used for pig samples induced a significant increase in the percentage of mobile sperm (82.3% and 76.4% after and before selection, respectively) and improved sperm morphological characteristics [32]. This highlights the importance of optimizing an ideal protocol for each species, since even closed phylogenetic species show different responses.

Before any processing, collared peccary semen already had excellent quality parameters. Nevertheless, it was still necessary to remove the SP for IVF and optimize the separation of the highest possible percentage of mobile spermatozoa with better kinetic patterns [17]. The results demonstrated the effective removal of SP and recovery of good kinetic patterns (motility) of collared peccary sperm using PG 45–90%. Additionally, sperm separation with colloid density (Porcicoll) contributed to the removal of microorganisms from swine semen samples [48]. Fresh semen from collared peccary is also contaminated with bacteria (2.3 ± 0.9 × 10^6^ colony-forming units/mL), which can impair sperm quality [49]. A method that allows for the separation of microorganisms from spermatozoa may benefit the use of collared peccary semen in techniques such as IVF.

Under the conditions studied in the present study, the swim-up method did not effectively support the isolation of mobile sperm from collared peccaries. Nevertheless, swim-up was successfully used to isolate highly mobile sperm in pigs [44]. Moreover, this method, when used to remove diluent from frozen/thawed pig semen after centrifugation, showed higher viability (89.3% from 79.7%) and a normal morphology (98.5% from 82.9%) compared with the pre-separation semen sample [20]. Moreover, this technique enhanced sperm motility in cattle (83.8% vs. 69.7%) [17] and yak (76.7% vs. 32.4%) [50] compared with unselected samples. In contrast, in llama (14.5% vs. 32.2%) [51] and goat (41.1% vs. 55.5%) [25], the swim-up method rather than Percoll^®^-based techniques showed the low recovery of mobile sperm, which is similar to the results observed in collared peccaries.

Considering the preparation of the collared peccary samples for swim-up, it was inferred that these sperm would be more sensitive to the physical stress caused by centrifugation than to the oxidative stress caused by SP. In the centrifuged samples, the functional integrity of the membrane and viability decreased, while the levels of ROS and the antioxidant defense of GSH increased. However, SP that was retained in the sperm sample caused high ROS-stimulated oxidative stress but did not alter the sperm quality. Castelo et al. [6] reported reduced sperm motility and increased morphological defects on centrifugation (750× *g*/3 min) before or after the semen underwent freeze/thawing.

Considering the inefficiency of the swim-up strategy, heterologous IVF was conducted using only the PG 45–90% group. This study demonstrates that heterologous IVF performed between collared peccary sperm and porcine oocytes obtains embryos developed up to the cleavage and morula stages. This technique for assessing sperm fertility could be crucial during the limited availability of oocytes, which is common in wild species [52]. Campos et al. [53] evaluated the binding ability of collared peccary sperm using immature porcine oocytes. In this study, it was observed that porcine oocytes could be used to assess the functionality of collared peccary spermatozoa, but the potential for fertilization and development of hybrid embryos has not been verified so far.

Before IVF, the PG group had a higher percentage of ALH, STR, and LIN than the WC group. Previously, in the collared peccary, the number of sperm bonded to porcine oocytes was positively correlated with STR; similar correlations were detected between LIN and sperm binding to the egg perivitelline membrane [53]. However, PG rather than WC increased the subpopulation of static sperm, which imposed a negative correlation in the egg perivitelline membrane binding test [53].

Therefore, collared peccary sperm separation techniques had no significant influence on sperm–oocyte interaction, fertilization, or the development of hybrid embryos. In swine species, PG allowed for a better penetration rate (96.2 %vs. 62.3%), cleavage (43.5% vs. 26.6%), and blastocyst (18.1% vs. 10.5%) compared to WC [54]. In cattle, a better blastocyst rate was also observed (56% vs. 38%) after sperm separation with PG compared to after washing [55]. Obtaining blastocyst-stage embryos after heterologous IVF of collared peccary sperm and pig oocytes was not feasible, likely due to the species differences and the variations in culture conditions.

Despite its simplicity, WC allowed for the same rate of fertilization and embryo development as PG did. These results may be related to the IVF system, where the sperm face fewer difficulties on the way to the oocyte compared to the female reproductive tract [43]. Differences between the PG and WC methods could be more evident if fertility is evaluated in vivo. Moreover, it is important to further evaluate the effects of these separation methods on frozen/thawed samples that show a loss of motility and quality during the process and may present unique patterns [56].

## 5. Conclusions

The 45–90% Percoll^®^ gradient was detected to be the most suitable one for collared peccaries. However, swim-up using the tested conditions did not yield satisfactory results and is not recommended for this species. According to the results of heterologous IVF, both PG 45–90% and WC can be used to remove SP and select sperm from semen samples of collared peccaries. Furthermore, successful fertilization and early embryonic development were achieved through heterologous IVF using collared peccary sperm and pig oocytes, providing a valuable alternative method for assessing the fertility of these animals.

## Figures and Tables

**Figure 1 animals-13-01955-f001:**
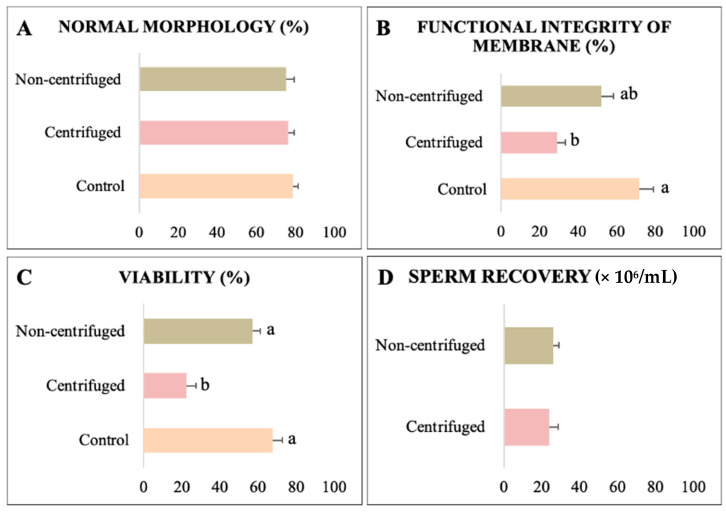
Quality and sperm recovery of collared peccary sperm separated by swim-up technique. (**A**) Normal morphology. (**B**) Functional integrity of the membrane. (**C**) Viability. (**D**) Sperm recovery. ^a,b^: values with different superscript letters are significantly different (*p* < 0.05).

**Figure 2 animals-13-01955-f002:**
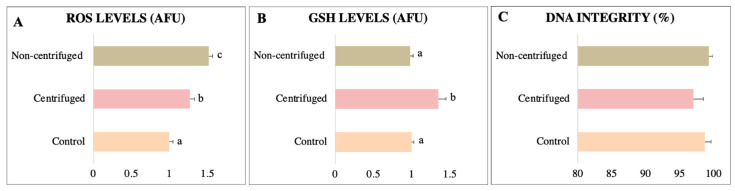
Physiological assessment of collared peccary sperm separated by the swim-up technique. (**A**) ROS levels. (**B**) GSH levels. (**C**) DNA integrity. ^a,b,c^: values with different superscript letters are significantly different (*p* < 0.05).

**Figure 3 animals-13-01955-f003:**
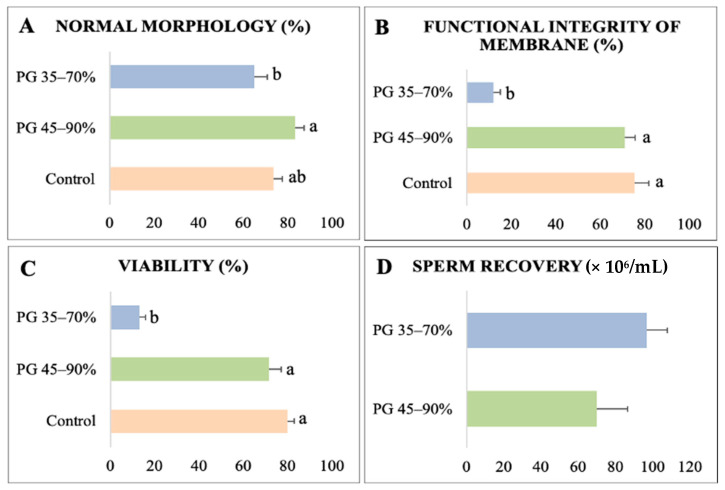
Quality and sperm recovery of collared peccary sperm separated by Percoll^®^ gradient technique. (**A**) Normal morphology. (**B**) Functional integrity of the membrane. (**C**) Viability. (**D**) Sperm recovery. ^a,b^: values with different superscript letters are significantly different (*p* < 0.05).

**Figure 4 animals-13-01955-f004:**
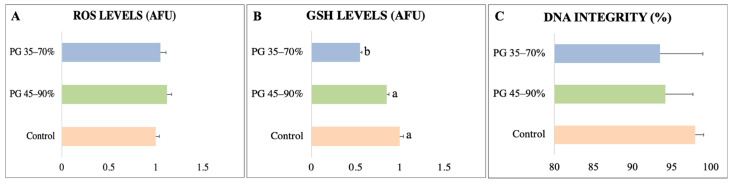
Physiological assessment of collared peccary sperm separated by Percoll^®^ gradient technique. (**A**) ROS levels. (**B**) GSH levels. (**C**) DNA integrity. ^a,b^: values with different superscript letters are significantly different (*p* < 0.05).

**Figure 5 animals-13-01955-f005:**
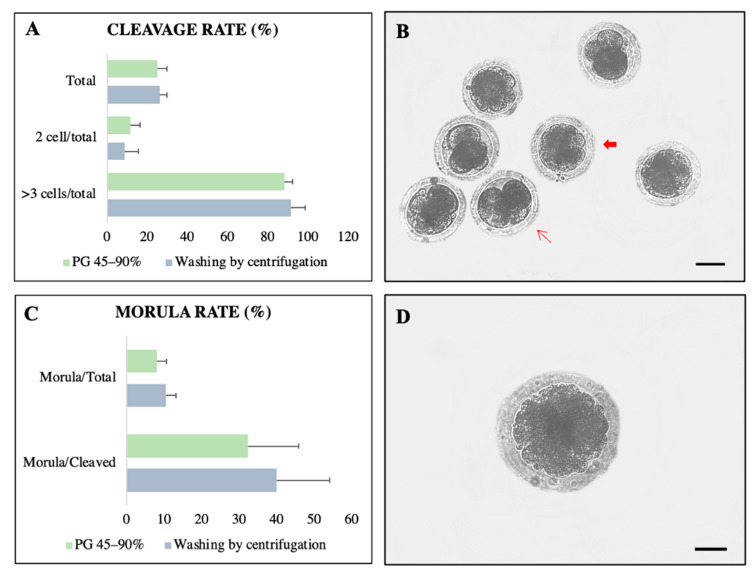
Embryonic development after heterologous fertilization of collared peccary sperm separated by PG 45–90% (*n* = 135 oocytes) and washing by centrifugation (*n* = 133 oocytes). (**A**) Percentage of embryos in D3 with two cells and three to seven cells. (**B**) Embryos two days after in vitro fertilization (thin arrow: embryo with three cells; wide arrow: embryo with more than three cells). (**C**) Percentage of embryos in D6. (**D**) Morula stage embryo after 6 days of development. No differences were observed among treatments (*p* > 0.05). Mean ± standard error. Six repetitions. Scale bars = 150 µm.

**Table 1 animals-13-01955-t001:** Motility patterns of collared peccary sperm separated by swim-up after the removal or not of seminal plasma (SP) by centrifugation.

CASA	Control (No Separation)	Swim-Up	
		Centrifuged (without SP)	Non-Centrifuged (with SP)
Motility (%)	90.4 ± 2.4 ^a^	31.4 ± 8.1 ^b^	51.9 ± 6.4 ^b^
Progressive motile (%)	68.6 ± 3.9 ^a^	17.7 ± 3.8 ^b^	28.9 ± 3.8 ^b^
Velocity average pathway (VAP; µm/s)	74.0 ± 4.5 ^a^	47.2 ± 5.1 ^b^	47.9 ± 1.9 ^b^
Velocity straight line (VSL; μm/s)	54.2 ± 6.1 ^a^	37.6 ± 4.3 ^b^	37.6 ± 2.0 ^b^
Curvilinear velocity (VCL; μm/s)	137.6 ± 5.3 ^a^	91.1 ± 9.1 ^b^	88.4 ± 3.5 ^b^
Amplitude of lateral head (ALH; μm)	6.1 ± 0.2 ^a^	5.1 ± 0.2 ^b^	4.8 ± 0.3 ^b^
Beat cross frequency (BCF; Hz)	36.3 ± 0.5 ^a^	35.3 ± 1.4 ^a^	35.9 ± 1.0 ^a^
Straightness (STR; %)	74.0 ± 3.1 ^a^	74.7 ± 1.5 ^a^	73.4 ± 2.3 ^a^
Linearity (LIN; %)	42.3 ± 3.2 ^a^	43.7 ± 1.8 ^a^	43.3 ± 2.5 ^a^
Rapid (%)	80.1 ± 2.9 ^a^	21.1 ± 5.2 ^b^	35.1 ± 4.2 ^b^
Medium (%)	9.9 ± 1.3 ^a^	10.3 ± 3.8 ^a^	17.1 ± 2.6 ^a^
Slow (%)	2.7 ± 0.6 ^a^	6.1 ± 1.4 ^a^	6.1 ± 1.8 ^a^
Static (%)	7.0 ± 2.2 ^a^	62.1 ± 8.6 ^b^	41.7 ± 8.0 ^b^

^a,b^: values (mean ± standard error) with different superscript letters within lines are significantly different (*p <* 0.05). Seven repetitions.

**Table 2 animals-13-01955-t002:** Motility patterns of collared peccary sperm selected by Percoll^®^ gradient at different densities.

CASA	Control (No Separation)	Percoll^®^ Gradient	
		PG 45–90%	PG 35–70%
Motility (%)	95.0 ± 1.8 ^a^	87.5 ± 3.6 ^a^	8.2 ± 2.1 ^b^
Progressive motile (%)	70.8 ± 6.0 ^a^	75.3 ± 5.2 ^a^	3.3 ± 1.4 ^b^
Velocity average pathway (VAP; µm/s)	69.7 ± 7.5 ^b^	96.8 ± 9.4 ^a^	40.7 ± 6.1 ^b^
Velocity straight line (VSL; μm/s)	56.0 ± 6.6 ^b^	88.4 ± 8.9 ^a^	27.3 ± 7.0 ^b^
Curvilinear velocity (VCL; μm/s)	121.4 ± 11.4 ^a^	129.6 ±7.5 ^a^	95.4 ± 12.0 ^a^
Amplitude of lateral head (ALH; μm)	5.4 ± 0.4 ^a^	4.4 ± 0.3 ^a^	6.2 ± 0.6 ^b^
Beat cross frequency (BCF; Hz)	36.4 ± 0.9 ^a^	36.0 ± 1.9 ^a^	38.8 ± 2.2 ^a^
Straightness (STR; %)	76.0 ± 2.2 ^a^	87.2 ± 1.4 ^a^	64.7 ± 6.2 ^b^
Linearity (LIN; %)	46.0 ± 3.1 ^b^	66.0 ± 4.7 ^a^	35.7 ± 6.7 ^b^
Rapid (%)	81.0 ± 6.2 ^a^	79.7 ± 5.3 ^a^	5.2 ± 1.6 ^b^
Medium (%)	14.0 ± 4.9 ^a^	7.7 ± 1.8 ^ab^	2.8 ± 0.7 ^b^
Slow (%)	2.2 ± 0.3 ^a^	2.7 ± 0.7 ^a^	2.3 ± 0.6 ^a^
Static (%)	3.0 ± 1.6 ^a^	10.0 ± 3.0 ^b^	89.8 ± 2.6 ^c^

^a,b,c^: values (mean ± standard error) with different superscript letters within lines are significantly different (*p <* 0.05). Six repetitions.

**Table 3 animals-13-01955-t003:** Motility patterns and morphology of the collared peccary sperm separated using Percoll^®^ gradient and washing by centrifugation for heterologous IVF.

CASA	Fresh Control	Percoll^®^ Gradient	Washing by Centrifugation
Motility (%)	96.8 ± 2.0 ^a^	66.3 ± 10.2 ^a^	89.0 ± 5.4 ^a^
Progressive motile (%)	59.5 ± 12.1 ^a^	47.5 ± 7.9 ^a^	56.0 ± 12.5 ^a^
Velocity average pathway (VAP; µm/s)	83.0 ± 8.7 ^a^	105.3 ± 7.3 ^a^	96.0 ± 8.8 ^a^
Velocity straight line (VSL; μm/s)	61.7 ± 9.3 ^b^	98.7 ± 7.2 ^a^	72.6 ± 9.3 ^ab^
Curvilinear velocity (VCL; μm/s)	157.3 ± 9.1 ^a^	139.4 ± 10.6 ^a^	166.7 ± 9.3 ^a^
Amplitude of lateral head (ALH; μm)	7.3 ± 0.2 ^b^	4.2 ± 0.6 ^a^	7.2 ± 0.6 ^b^
Beat cross frequency (BCF; Hz)	34.4 ± 0.7 ^a^	36.2 ± 2.5 ^a^	33.1 ± 1.7 ^a^
Straightness (STR; %)	71.3 ± 3.7 ^b^	92.0 ± 3.2 ^a^	72.5 ± 3.6 ^b^
Linearity (LIN; %)	40.3 ± 3.9 ^b^	72.3 ± 4.7 ^a^	46.0 ± 4.7 ^b^
Rapid (%)	62.5 ± 12.3 ^a^	48.8 ± 8.5 ^a^	59.5 ± 12.6 ^a^
Medium (%)	11.8 ± 4.0 ^a^	4.0 ± 4.0 ^a^	7.0 ± 1.5 ^a^
Slow (%)	22.3 ± 6.7 ^a^	8.5 ± 2.7 ^a^	22.3 ± 6.2 ^a^
Static (%)	3.3 ± 2.0 ^a^	39.0 ± 11.2 ^b^	11.0 ± 5.4 ^a^
Normal morphology (%)	79.7 ± 7.4 ^a^	82.0 ± 6.4 ^a^	82.0 ± 6.4 ^a^

^a,b^: values (mean ± standard error) with different superscript letters within lines are significantly different (*p <* 0.05). Four repetitions.

**Table 4 animals-13-01955-t004:** Sperm–oocyte interaction and fertilization after heterologous IVF of collared peccary sperm separated by different techniques.

Selection Technique	Matured Oocytes, %	Sperm Bound to Matured Oocytes	Sperm Bound to Non-Matured Oocytes	Sperm Bound to Total Oocytes	Second Polar Body, %	Two Pronuclei, %	Fertilized Oocytes	
							Total, %	Matured, %
Percoll gradient	65.6 ± 1.2 (40/61)	4.9 ± 2.8 (196/40)	9.3 ± 4.3 (196/21)	3.2 ± 1.6 (196/61)	9.8 ± 3.1 (6/61)	4.9 ± 2.7 (3/61)	14.8 ± 4.9 (9/61)	22.5 ± 10.8 (9/40)
Washing by centrifugation	52.5 ± 1.5 (32/61)	3.2 ± 1.9 (103/32)	3.6 ± 4.7 (103/29)	1.7 ± 1.3 (103/61)	3.3 ± 2.4 (2/61)	9.8 ± 3.6 (6/61)	13.1 ± 4.0 (8/61)	25.0 ± 9.5 (8/32)

No differences were observed among treatments (mean ± standard error, *p* > 0.05). Six repetitions.

## Data Availability

The data presented in this study are available upon request from the corresponding author.

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
