# Peer review of "Different Methods for Seminal Plasma Removal and Sperm Selection on the Quality and Fertility of Collared Peccary Sperm"

_animals, 2023, doi:10.3390/ani13121955_

Round 1

Reviewer 1 Report

The authors compared several methods including the swim-up, Percoll® gradient (PG) and washing by centrifugation (WC), for the application of IVF in collared peccary, a important species for maintaining the ecosystem in America. The experimental design, methodology, data collection and interpretation are all nice. The manuscript is organized well and written in fluent English.

The reviewer noticed that there is a literature closely related to this topic was published most recently in Animals, but it's not cited in the references of this manuscript.

1. Effect of Detergents Based on Sodium Dodecyl Sulfate on Functional Metrics of Frozen–Thawed Collared Peccary (Pecari tajacu) Semen. Animals 2023, 13(3), 451

For Table 1, it's better to put the full names of those abbreviations following the table for the convenience of readers.

Renew the references (including the most recently published articles closely related to your topic).

For Fig 5, usualy embryos of 2-cell, 4-cell, 8-cell, morula and blastocyst were investigated. Why 3-cell embryos in your data? What about the situation of  4-cell, 8-cell, and blastocyst embryos?

The quality of English expression is nice.

Author Response

Response to comments from Reviewer #1:

The authors compared several methods including the swim-up, Percoll® gradient (PG) and washing by centrifugation (WC), for the application of IVF in collared peccary, an important species for maintaining the ecosystem in America. The experimental design, methodology, data collection and interpretation are all nice. The manuscript is organized well and written in fluent English. The reviewer noticed that there is a literature closely related to this topic was published most recently in Animals, but it's not cited in the references of this manuscript. Effect of Detergents Based on Sodium Dodecyl Sulfate on Functional Metrics of Frozen–Thawed Collared Peccary (Pecari tajacu) Semen. Animals 2023, 13(3), 451

Authors: Initially, we are grateful for the attention and suggestions made by the reviewer. The reference was added in the manuscript (Page 2, Line 60).

For Table 1, it's better to put the full names of those abbreviations following the table for the convenience of readers.

Authors: The full names were added in the Table 1 (Pages 6–7, Lines 282–285). We also made this correction in Table 2 (Pages 8–9, Lines 319–321) and Table 3 (Page 10, Lines 348–351).

Renew the references (including the most recently published articles closely related to your topic).

Authors: We added some new references in the manuscript with publications from 2022 and 2023.

For Fig 5, usually embryos of 2-cell, 4-cell, 8-cell, morula, and blastocyst were investigated. Why 3-cell embryos in your data? What about the situation of 4-cell, 8-cell, and blastocyst embryos?

Authors: We appreciate the reviewer's comment. A low number of embryos were acquired using heterologous IVF, and the cleaved embryos were classified in 2-cell and 3 or more cells after 48 h of culture. They were grouped this way to show data about how many embryos were in the first division (2-cell) or second and following divisions.  Regarding to blastocysts, it was not possible to obtain embryos at this stage after heterologous IVF of collared peccary spermatozoa and porcine oocytes, possibly due to the difference between species and culture conditions. We describe this information in the manuscript (Page 13, Lines 455–457).

Reviewer 2 Report

With my best regards, 

The authors are prized in the selection of the topic, which is bears fundamental importance. 

Although the topic of the study raise the curiosity of this reviewer, the present state of the submitted work still lacks refinement, clarification and organization.

Both summary and abstract lack any numerical display of results or statistic information, are confusing in introducing the topic and in presenting the results and conclusion.

I would recommend an overall re-writing of the introduction, there are several mistakes and confusing statements. 

Line 51 - there's a word missing after ''has been''

Lines 74-81 - There's clear inconsistencies regarding the utilization of centrifugation, first describing as negative for sperm quality, then indicating its utilization as alternative in selection methods.

Line 85-86 - A description of the method should be performed.

Line 93 - change ''is'' for ''has been''

*Material and methods

There's a very unclear description of the methodology, including the type of sperm sample used in each one of the 3 experiments. Is it one sample? The samples were frozen? Were there different collections or samples from the same animal?

***The utilization of first person''we'' was observed several times, and in scientific writing, such perspective is not a match.

The first paragraph of the semen collection section is very confusing and should be rewritten. 

In addition, the experiment 3 does not include non-selected sperm, which displayed a high quality before being subjected to the selection methods, and in an overall perspective, it's a major flaw for this reviewer. 

Line 171-173 - It's unclear what was performed first

*Statistical analyzes section

This section raised my curiosity in association with the results; A further description why the data does not follow a normal distribution is required, as well as why chy-squared test was performed and for which parameters it was used.

*Results

Figure 2 and 4 are the same, there was a mistake in adding the correct figure. 

Very high SEM observed in several parameters; identification of possible outliers could benefit the overall analyzes of the results.

**Interestingly, the same Percoll method, performed significantly worse when comparing the results from Table 2 and 3, what's the reason for it?

*Discussion

Line 373  - Indicated that BAP, VSL and LIN were higher, however on table 3, the difference is non-significant. 

The discussion still lacks focus on the major findings of the study, and in few sections, focus on non-interesting information for the study.

*Conclusion

Line 462 - The sentence starting with Furthermore and continuing to line 463 does not relate to anything stated in the aims or objectives of the study and therefore should not be in the conclusion.

I hope the modifications indicated may assist in the reformulation of the manuscript.

Best regards.

The overall response regarding the study is that different sections have complete different writing style, whereas no issues were observed in material and methods and discussion/conclusion, the introduction is sub-par, very confusing and unclear. A thorough revision of the english language is recommended. 

Author Response

Response to comments from Reviewer #2:

The overall response regarding the study is that different sections have complete different writing style, whereas no issues were observed in material and methods and discussion/conclusion, the introduction is sub-par, very confusing and unclear. A thorough revision of the english language is recommended. The authors are prized in the selection of the topic, which is bears fundamental importance.

Authors: Initially, we are grateful for the attention and suggestions made by the reviewer. We appreciate your comments. The entire manuscript was reviewed to clarifies the ideas, especially the introduction as recommended. The manuscript version had been sent to a company specializing in English-language manuscript review (certificate of proof attached to submission). Now, we once again submit it to the company for a second expert review (document attached to the manuscript submission). Finally, we carefully reviewed the entire manuscript, with the aim of presenting it in a concise, clear manner, with the reviewer's suggestions included.

Although the topic of the study raise the curiosity of this reviewer, the present state of the submitted work still lacks refinement, clarification and organization.

Authors: The entire manuscript was reviewed, and the corrections were performed.

Both summary and abstract lack any numerical display of results or statistic information, are confusing in introducing the topic and in presenting the results and conclusion.

Authors: We appreciate the reviewer's comment. We have reorganized both the simple summary and abstract with the suggestions made by the reviewer and within the rules of the journal Animals. (Page 1).

I would recommend an overall re-writing of the introduction, there are several mistakes and confusing statements.

Authors: This topic was carefully reviewed (Page 2). Moreover, we check the grammatical writing of the information with an English language professional, which issued the English language editing certificate (document attached to the manuscript submission).

Line 51 – There's a word missing after ''has been''

Authors: The sentence was corrected (Page 2, Lines 51–52).

Lines 74-81 – There's clear inconsistencies regarding the utilization of centrifugation, first describing as negative for sperm quality, then indicating its utilization as alternative in selection methods.

Authors: We agree with the reviewer's comment. In fact, the way in which the information was presented was confusing as to the proposition of the manuscript. In a first study carried out with spermatozoa centrifugation of collared peccaries, Castelo et al. used a high rotation for the centrifugation of cryopreserved collared peccary spermatozoa. Now, in this present study, we evaluated centrifugation on fresh spermatozoa by reducing this centrifugation rotation; however, observing the formation of the pellet after centrifugation. This problematization was better presented in the introduction. We referred to high intensity centrifugation and corrected the manuscript (Page 2, Lines 74–94).

Line 85-86 – A description of the method should be performed.

Authors: The description was added in the manuscript (Page 2, Lines 85–86).

Line 93 – change ''is'' for ''has been''

Authors: The text was corrected (Page 2, Line 93).

*Material and methods

There's a very unclear description of the methodology, including the type of sperm sample used in each one of the 3 experiments. Is it one sample? The samples were frozen? Were there different collections or samples from the same animal?

Authors: The methodology was clarified with all these information (Pages 3–4, Lines 108–109, 128–133, 155–157, 171–172). The samples were fresh from 19 animals divided in experiment I (7 animals), II (6 animals) and III (6 animals). The semen from each animal was considered as one replicate and divided between experimental groups described on the experimental design.

***The utilization of first person''we'' was observed several times, and in scientific writing, such perspective is not a match.

Authors: The text was corrected in the manuscript.

The first paragraph of the semen collection section is very confusing and should be rewritten.

Authors: The text was corrected in the manuscript (Page 3, Lines 108–119).

In addition, the experiment 3 does not include non-selected sperm, which displayed a high quality before being subjected to the selection methods, and in an overall perspective, it's a major flaw for this reviewer.

Authors: We appreciate the reviewer's comment. As mentioned in the introduction (Page 2, Lines 66–73), to perform IVF with fresh semen it is necessary to remove the seminal plasma, for this reason it is not possible to perform the technique using non-selected sperm and sperm in suspension with seminal plasma.

Line 171-173 – It's unclear what was performed first

Authors: The text was corrected (Page 4, Lines 163–174).

*Statistical analyzes section: This section raised my curiosity in association with the results; A further description why the data does not follow a normal distribution is required, as well as why chy-squared test was performed and for which parameters it was used.

Authors: The paragraph was corrected. Chi-squared test for categorical variables independent of each other was used to compare sperm-oocyte interaction, fertilization, cleavage rate and morula rate (Page 6, Lines 265–272).

*Results

Figure 2 and 4 are the same, there was a mistake in adding the correct figure.

Authors: We apologize for the error. The figure was corrected (Page 8, Line 335).

Very high SEM observed in several parameters; identification of possible outliers could benefit the overall analyzes of the results.

Authors: Thank you for the advice. We had verified this previously, the replicates presented are the most homogeneous that we had.

**Interestingly, the same Percoll method, performed significantly worse when comparing the results from Table 2 and 3, what's the reason for it?

Authors: This result is probably due differences between the animals used, as they were not the same animals in the two experiments. In collared peccaries, Peixoto et al. (2012) have observed individual variations in the seminal characteristics of this species.

- Peixoto, G.C.X, Silva, M.A., Castelo, T.S., Silva, A.M., Bezerra, J.A.B, Souza, A.L.P., Oliveira, M.F., Silva, A.R. Individual variation related to testicular biometry and semen characteristics in collared peccaries (Tayassu Tajacu Linnaeus, 1758). Animal Reproduction Science, 2012, 134, 191– 196.

*Discussion

Line 373 – Indicated that BAP, VSL and LIN were higher, however on table 3, the difference is non-significant.

Authors: The text was corrected since we referred to Table 2 (Page 12, Lines 375–376).

The discussion still lacks focus on the major findings of the study, and in few sections, focus on non-interesting information for the study.

Authors: The text was reviewed (Pages 12–13). We revised the entire discussion, inserted new information, and rearranged the information according to reviewers' comments.

*Conclusion

Line 462 – The sentence starting with Furthermore and continuing to line 463 does not relate to anything stated in the aims or objectives of the study and therefore should not be in the conclusion.

Authors: The sentence has been rewritten (Page 13, Lines 470–472).

I hope the modifications indicated may assist in the reformulation of the manuscript.

Authors: We appreciate the reviewer's careful review and have made all suggested changes.

Round 2

Reviewer 2 Report

No other comments from this reviewer. The modifications performed by the authors as well as the questions answered were satisfactory.